# HER2-Positive Metastatic Breast Cancer: Available Treatments and Current Developments

**DOI:** 10.3390/cancers15061738

**Published:** 2023-03-13

**Authors:** Ismail Essadi, Zineb Benbrahim, Mohamed Kaakoua, Thibaut Reverdy, Pauline Corbaux, Gilles Freyer

**Affiliations:** 1Medical Oncology, Ibn Sina Military Hospital, Faculty of Medicine, Cadi Ayyad University, Marrakesh 40080, Morocco; 2Medical Oncology, Faculty of Medicine, Sidi Mohamed Benabdellah University, Fez 30000, Morocco; 3Medical Oncology, HCL Cancer Institute, 69310 Lyon, France; 4Medical Oncology, University Hospital Center, 42000 Saint-Etienne, France

**Keywords:** metastatic breast cancer, HER2-positive, targeted therapies

## Abstract

**Simple Summary:**

Since the advent of trastuzumab in HER2-positive metastatic breast cancer management, the natural history of this disease continues to improve. The development of molecular biology and better knowledge of the resistance mechanisms of cancer cells have enabled the development of several therapies targeting HER2 and consequently improved the overall survival of these patients. This paper aims to review the new therapies developed for this entity of breast cancer, their tolerance profiles, and their positions in therapeutic strategy.

**Abstract:**

For several years, the overexpression of the HER2 receptor in breast cancer has been correlated with a poor prognosis and an increased risk of developing brain metastases. Currently, the combination of anti-HER2 double blockade and taxane and trastuzumab emtansine (T-DM1) are considered the standard treatments for metastatic breast cancer overexpressing these receptors in the first and second line. Very recently, the development of a new antidrug conjugate, trastuzumab–deruxtecan, has improved the overall survival of patients, even in second-line treatment. However, trastuzumab–deruxtecan has become a new standard. Despite the benefits of these antidrug conjugates, this benefit in patients with brain metastases remains unclear. Tucatinib is a new tyrosine kinase inhibitor that has given hope for the treatment of these patients. The objective of this article was to review data on the established drugs and novel agents for HER2-positive MBC and to discuss how to incorporate anti-HER2 therapies in first and later-line settings.

## 1. Introduction

Breast cancer is the leading cause of cancer-related mortality among females in the world [1]. Approximately 15 percent of breast cancers overexpress human epidermal growth factor receptor 2 (HER2) [2]. HER2-positive breast cancer is diagnosed by IHC. A strong expression in IHC (score 3+) is sufficient. For the equivocal forms (score 2+), a recourse to in-situ hybridization (ISH) is necessary to rule on the amplification of the HER2 gene. The scores 1+ and 0 are systematically considered as negative [3]. The management of HER2-positive metastatic breast cancer (MBC) has dramatically changed over the last decade from an aggressive disease with a poor prognosis to a chronic disease with prolonged survival [4]. This is related to a better understanding of HER2 biology leading to the development of highly effective targeted therapies. An accurate determination of HER2 status is important to maximize the benefit and avoid unnecessary toxicity. Just a while ago, only cancers that have a test score of 3+ staining by immunohistochemistry for the HER2 protein or HER2 gene amplification by fluorescence in situ hybridization respond to HER2-targeted agents. However, with the second-generation ADCs, a new era has opened up where efficacy can be seen even on “low” HER2-expressing tumors (score 1+ and 2+ with negative ISH) [5,6]. In early studies with trastuzumab, heart failure was identified as a major adverse effect of this agent, especially when co-administered with anthracyclines. Subsequently, the systematic evaluation of cardiac function in the studies made it possible to better understand the nature of the cardiotoxicity induced by the various anti-HER2 therapies. This cardiotoxicity (involving all the anti-HER2 molecules), being reversible, requires regular monitoring of the ejection fraction of the left ventricle estimated by echocardiography [4]. The objective of this article was to review the data on established drugs and novel agents for HER2-positive MBC and to discuss how to incorporate anti-HER2 therapies in first- and later-line settings.

## 2. First Line Treatment

### 2.1. Trastuzumab–Pertuzumab

Trastuzumab is a humanized monoclonal antibody against subdomain IV of HER2. The addition of trastuzumab to chemotherapy improves objective response rates, progression-free survival (PFS), and overall survival (OS) in HER2-positive patients with MBC [7]. In a meta-analysis involving 1497 patients with HER2-positive MBC, trastuzumab extended the overall survival by 5 to 8 months (hazard ratio (HR): 0.82, 95% CI 0.71 to 0.94) [8]. Although it also increased the risk of congestive heart failure (HR: 3.49, 90% CI 1.88 to 6.47), cardiotoxicity was reversible with holding treatment. In a recent meta-analysis, using trastuzumab was reported as an important risk factor for cardiotoxicity observed in patients initially treated with anthracyclines (OR: 1.45, 95% CI: 1.28–1.65, *p* < 0.00001) [9], Hence the importance of maintaining a high level of monitoring.

Pertuzumab inhibits the dimerization of HER2 with other HER receptors (main pathway of resistance to trastuzumab). Therefore, the combination of trastuzumab with pertuzumab leads to a complimentary blockade of HER2 signaling [10]. Early data on pertuzumab efficacy informed the design of the CLEOPATRA trial, in which 808 newly diagnosed patients with HER2-positive MBC were randomly assigned to standard trastuzumab, docetaxel, and pertuzumab (THP) or trastuzumab, docetaxel and a placebo. Double blockade resulted in a 38% improvement in PFS (18.5 months vs. 12.4 months, *p* < 0.001) [11,12,13]. The median OS was 57.1 months in the THP arm vs. 40.8 months in the control arm (*p* < 0.001); 8-year landmark overall survival rates were 37% versus 23% for the experimental group. The safety profile of the THP combination arm was similar to the placebo arm. There was no increase in left ventricular systolic dysfunction (2% in the control group compared to the 1% in the pertuzumab group), but more febrile neutropenia was seen in the pertuzumab group (13.8% vs. 7.6), as well as grade 3 diarrhea (7.9% vs. 5.0%).

On the basis of these findings, the 5th ESO-ESMO guidelines for advanced breast cancer (ABC 5) and the NCCN guidelines consider THP as the standard of care for first-line treatment of HER2-positive MBC [14].

In the CLEOPATRA study, 88 patients who previously received anti-HER2 therapy with a free interval greater than 12 months were included. Therefore, we consider the combination of chemotherapy with pertuzumab and trastuzumab as an important first-line therapeutic option for this subset of patients [14].

Other chemotherapy backbones such as paclitaxel, nab-paclitaxel [15], and vinorelbine [16] may be substituted for docetaxel. The PERUSE study [15] compared docetaxel, paclitaxel, and nanoparticle albumin-bound paclitaxel, in combination with trastuzumab and pertuzumab with no random assignment, among 1436 patients with HER2-positive MBC. The median PFS were comparable (20, 23, and 18 months, respectively).

The association of endocrine therapy with trastuzumab and pertuzumab is another option for patients with postmenopausal HR and HER2-positive MBC. This is based on the results of the PERTAIN study testing pertuzumab plus trastuzumab and oral anastrozole or letrozole versus trastuzumab and an AI. This study concluded to increase PFS with the double blockade combination (18.9 and 15.8 months respectively; HR: 0.65, 95% CI 0.48 to 0.89) [17]. Endocrine therapy with double HER2 blockade is then an appropriate strategy, with low toxicity, for non-rapidly progressive postmenopausal HR and HER2-positive MBC or after chemoinduction [expert consensus].

Finally, single-agent trastuzumab or trastuzumab plus pertuzumab may also be an option for patients who want to avoid cytotoxic chemotherapy [18,19,20].

Research manuscripts reporting large datasets that are deposited in a publicly available database should specify where the data have been deposited and provide the relevant accession numbers. If the accession numbers have not yet been obtained at the time of submission, please state that they will be provided during review. They must be provided prior to publication.

Interventionary studies involving animals or humans, and other studies that require ethical approval, must list the authority that provided the approval and the corresponding ethical approval code.

### 2.2. Ado-Trastuzumab Emtansine 

T-DM1 is an antibody-drug conjugate that links the chemotherapy drug emtansine to trastuzumab providing the targeted delivery of this drug to HER2-positive tumor cells [21]. Its efficacy has been mainly shown beyond first-line HER2-directed therapy and will be discussed further in this review. In the first-line setting, the MARIANNE trial compared TDM1 with or without pertuzumab to trastuzumab plus a taxane. Treatment arms containing T-DM1 showed a non-inferior PFS outcome compared with a control arm with no further benefit of pertuzumab over TDM1 alone [22]. Based on these findings, T-DM1 may be considered as an alternative in the first-line setting for patients who have had rapid disease progression (<6 months) following adjuvant therapy with trastuzumab (indication retained by the results observed in the EMILIA trial, which we will detail later). Other alternatives should be offered to patients who received adjuvant TDM-1 for a residual disease after double blockade HER2 therapy in the neoadjuvant setting in case of metastatic relapse [23].

## 3. Second Line Treatment

### 3.1. Trastuzuma–Pertuzumab

The phase III PHEREXA study included patients who experienced disease progression following trastuzumab-based therapy to be assigned to trastuzumab, plus capecitabine with or without pertuzumab. The primary end point was PFS, which was not significantly improved with the addition of pertuzumab (9.0 vs. 11.1 months; HR: 0.82; 95% CI 0.65 to 1.02; *p* = 0.0731) [24]. The final OS showed a 9.1-month increase in survival in the pertuzumab arm [25]. Based on these data, the ABC 5 guidelines claims the combination of chemotherapy with trastuzumab and pertuzumab as an acceptable option for HER2-positive MBC in progression following trastuzumab, only if the double blockade was not received previously [14]. 

### 3.2. Ado-Trastuzumab Emtansine

The EMILIA trial enrolled 978 HER2-positive breast cancer patients previously treated with trastuzumab and a taxane. The patients were randomly assigned to T-DM1 or the combination of capecitabine plus lapatinib, repeated every three weeks [26]. T-DM1 resulted in an improvement in PFS and OS compared with lapatinib and capecitabine (PFS, 10 vs. 6 months, respectively; HR: 0.65, 95% CI 0.55 to 0.77 and OS, 31 vs. 25 months; HR: 0.68, 95% CI 0.55 to 0.85 respectively). Although a permitted crossover, a subsequent report of a longer follow-up confirmed the improved survival with T-DM1 [27]. TDM-1 significantly improved the overall response rate (44 vs. 31 percent) with a lower rate of serious toxicity overall (41 vs. 57 percent), including diarrhea, palmar plantar erythrodysesthesia, and vomiting. The most common serious toxicities associated with T-DM1 were thrombocytopenia and elevated liver function enzymes. Based on this success, T-DM1 was considered as the preferred option in patients who have progressed through at least one line of trastuzumab-based therapy. The activity of T-DM1 following progression on pertuzumab has been suggested in a retrospective study where one-third of patients were treated for more than 6 months with T-DM1 [28].

## 4. Third Line and Beyond

### 4.1. Previously Established Options

#### 4.1.1. Lapatinib

Lapatinib is tyrosine kinase inhibitor (TKI) that reversibly inhibits the intracellular tyrosine kinase domains of both epidermal growth factor receptor (EGFR) and HER. The efficacy of lapatinib plus capecitabine was found in a phase III trial of 399 patients randomized to receive lapatinib plus capecitabine or capecitabine alone [29,30,31]. Compared with capecitabine alone, the lapatinib plus capecitabine combination showed a higher time to tumor progression (median, 6 vs. 4 months) with a trend towards an improvement in THE OS (median, 75 vs. 65 weeks, not statistically significant). 

#### 4.1.2. Neratinib

Neratinib is an orally available irreversible inhibitor of the HER (Erb) TK domain and P-glycoprotein [32]. Compared to lapatinib, neratinib is characterized by a lower molecular weight. The NALA study was designed to include 621 patients with metastatic HER2-positive breast cancer who had received two or more prior anti-HER2-based regimens. Those patients were randomized to receive neratinib and capecitabine or lapatinib and capecitabine. The neratinib arm improved a mean PFS relative to the lapatinib arm (8.8 vs. 6.6 months, respectively; HR: 0.76, 95% CI 0.63 to 0.93), although OS results were similar (mean OS of 24 vs. 22 months, respectively; HR: 0.88, 95% CI 0.72 to 1.07) [33]. The most common grade ≥3 adverse event was diarrhea (24% vs. 13%). Patients with hormone receptor–negative disease derived the greatest PFS benefit from neratinib. This exploratory subgroup analysis may illustrate the bidirectional crosstalk between HER2 and estrogen-receptor pathways with estrogen-receptor signaling activation in case of inhibition of HER2 alone [33]. Fewer interventions for CNS disease occurred with neratinib than lapatinib (22.8% vs. 29.2%; *p* = 0.043). Of note, in a previous phase III (NEfERT-T study), 479 women with previously untreated HER2-positive MBC were randomized to receive neratinib versus trastuzumab combined with paclitaxel [34]. Both combinations showed a similar median PFS of 12.9 months (the primary objective of the study). Neratinib showed benefits in terms of the incidence of central nervous system recurrences (HR: 0.48, 95% CI 0.29 to 0.79; *p* = 0.002) and delayed time to central nervous system metastases (HR: 0.45, 95% CI 0.26 to 0.78; *p* = 0.004).

Therefore, both lapatinib and neratinib are appropriate options after ≥2 lines of HER2-directed treatment allowing the avoidance of infusional therapy and possible activity against central nervous system metastases. Taking into account the benefit of neratinib and capecitabine over lapatinib and capecitabine, it is preferable to use neratinib in this indication and revert to lapatinib plus capecitabine in case of gastro-intestinal toxicity. 

#### 4.1.3. Trastuzumab with Cytotoxic Agents 

The continuation of trastuzumab is common practice although there are low evidence data supporting this approach [18,35]. In one retrospective study which included 69 patients who were retreated with a trastuzumab-based strategy after disease progression on lapatinib, the ORR was 31% with a median duration of response of 8 months [36]. The median PFS and OS were 5 and 15 months, respectively. 

#### 4.1.4. Lapatinib–Trastuzumab Association

In one trial (EGF104900), lapatinib alone or in combination with trastuzumab was tested in 296 patients, with disease progression following one or more prior trastuzumab-containing regimens [37,38]. The combination resulted in higher PFS (11 vs. 8 weeks; HR: 0.74, 95% CI 0.58 to 0.94) and OS (14 vs. 10 months; HR: 0.74, 95% CI 0.57 to 0.97) compared to lapatinib alone. A more pronounced benefit in OS was seen after exclusion from the analysis of the population crossing from lapatinib alone to lapatinib–trastuzumab. The combination has also been documented as more effective in hormone receptor-negative populations. This led to more investigations in combination with endocrine therapy in HR and HER2-positive populations. The ALTERNATIVE trial included 355 postmenopausal women with HR and HER2-positive MBC previously treated with trastuzumab and prior endocrine therapy. Patients were randomly assigned to lapatinib plus trastuzumab plus an AI, lapatinib plus an AI, or trastuzumab plus an AI [39]. Lapatinib plus trastuzumab and an AI showed higher PFS and ORR relatively to AI plus trastuzumab without lapatinib (11 vs. 5.6 months; HR: 0.62, 95% CI 0.45 to 0.88 and 31.7 vs. 13.7 percent). This three-drug combination may be particularly attractive for patients who wish to avoid, or are not candidates for, chemotherapy.

### 4.2. Recently Approved Drugs

Resistance to trastuzumab emtansine may develop through the down-regulation or loss of HER2 expression, heterogeneous HER2 expression, receptor mutation, or expression of different HER2 isoform cancer cells may also develop resistance to the emtansine payload [40,41]. A comprehension of these mechanisms of resistance have led to the development of new drugs.

#### 4.2.1. Fam-Trastuzumab Deruxtecan

Fam-trastuzumab deruxtecan is an antibody-drug conjugate that includes a humanized HER2 antibody (trastuzumab) and a topoisomerase I inhibitor conjugate (deruxtecan) [42]. In a phase 1 study, trastuzumab deruxtecan was tested at a dose of 5.4 mg or 6.4 mg/kg in 111 patients with advanced HER2-positive breast cancer previously treated with trastuzumab emtansine [43]. The response rate was 59.5% (95% CI 49.7 to 68.7), and the median response duration was 20.7 months. On the basis of this study, the recommended dose of trastuzumab deruxtecan was 5.4 mg/kg. 

In a two-part, multicenter, phase 2 study (DESTINY Breast 01), trastuzumab deruxtecan was evaluated in patients with HER2-positive metastatic breast cancer previously treated with trastuzumab emtansine [44]. The primary end point of the study was the objective response rate. Trastuzumab deruxtecan provided a tumor response in 60.9% of patients (95% CI53.4 to 68.0) and durable antitumor activity with a median progression-free survival of 16.4 months (95% CI 12.7 to not reached). Efficacy results were consistent across sub-groups, including patients previously treated with pertuzumab. During the study, the most common adverse events of grade 3 or higher were neutropenia (20.7%), anemia (in 8.7%), and nausea (in 7.6%). Fam-trastuzumab deruxtecan was associated with interstitial lung disease in 13.6% of cases, including one patient who died from this complication. Patients receiving fam-trastuzumab deruxtecan should be monitored for any new respiratory symptoms and should be permanently discontinued if grade 2 or higher interstitial lung disease/pneumonitis develops.

These results were evaluated in a randomized phase 3 study (Destiny-Breast 02), comparing fam-trastuzumab deruxtecan with the treatment physician’s choice in patients with advanced HER2 breast cancer previously treated with trastuzumab emtansine [44]. After a median follow-up of 20 months, the median PFS is 17.8 months vs. 6.9 months in the fam-trastuzumab deruxtecan arm (HR 0.36, 95% CI 0.28–0.45, *p* < 0.00001). Benefits are also described in OS (39.2 vs. 26.5 months, respectively, HR 0.66, 95% CI 0.50–0.86, *p* = 0.0021) [45]. Trastuzumab deruxtecan continued to demonstrate these benefits, this time in the second line in patients with metastatic breast cancer previously treated with trastuzumab and taxane. The Destiny Breast 03 study (phase 3 trial) showed the superiority of trastuzumab deruxtecan over trastuzumab emtensine in progression-free survival and overall survival [45]. Median progression-free survival was 28.8 months with trastuzumab deruxtecan and 6.8 months with trastuzumab emtansine (HR: 0.33, 95% CI 0.26–0.43; *p* < 0.0001). Median overall survival was not achieved in either arm. The overall survival rate at 24 months was 77.4% in the trastuzumab deruxtecan arm compared to 69.9% in the trastuzumab emtensine arm [45]. Grade 3 adverse events were similar in both arms (56% in the trastuzumab deruxtecan arm versus 52%). In the experimental group, interstitial lung disease or drug-induced lung disease was noted in 15% versus 3% [46]. Based on these data, trastuzumab deruxtecan appears to be a new standard of care in the second-line setting.

#### 4.2.2. Tucatinib, Capecitabine, and Trastuzumab 

Tucatinib is an oral tyrosine kinase inhibitor which selectively binds to and inhibits the kinase domain of HER2 [47]. The efficacy of tucatinib has been documented through the HER2CLIMB trial including a total of 612 patients with HER2-positive MBC, previously treated with trastuzumab, pertuzumab, and trastuzumab emtansine [48]. A significant percentage of patients with untreated or previously treated progressing “active” brain metastases were included in the study. Patients were randomly assigned to receive tucatinib or a placebo, in combination with trastuzumab and capecitabine. The primary end point of the study was PFS. The tucatinib combination resulted in higher 1-year PFS (33.1% vs. 12.3%; HR: 0.54, 95% CI 0.42 to 0.71) and higher OS at 2 years (44.9% vs. 26.6%; HR: 0.66; 95% CI 0.50 to 0.88; *p* = 0.005) [47]. OS was longer by 4.5 months with tucatinib (median of 21.9 months vs. 17.4 months; HR: 0.66 (95% CI 0.50 to 0.88; *p* = 0.005)). Among patients with CNS metastases, the one-year PFS rate among the tucatinib combination arm was 24.9% vs. 0% in the placebo-combination arm (HR: 0.48, 95% CI 0.34 to 0.69; *p* < 0.001). Common adverse events observed in the tucatinib group included palmar-plantar erythrodysesthesia syndrome, diarrhea, vomiting, elevations in ALT and AST levels, and fatigue. Tucatinib plus trastuzumab and capecitabine is an active combination in heavily pretreated patients with HER2-positive metastatic breast cancer, including those with untreated or treated brain metastases [48].

#### 4.2.3. Margetuximab

Margetuximab is a chimeric Fc-engineered anti-HER2-receptor monoclonal antibody that shares epitope specificity and Fc-independent antiproliferative effects with trastuzumab [49].

The modified Fc domain of margetuximab improves its binding affinity to CD16A and decreases its binding affinity to CD32B, thus promoting its antitumor activity. The role of pharmacogenetic polymorphisms of CD16A on the efficacy of trastuzumab has been reported in two retrospective studies that suggest that patients with HER2-positive breast cancer with a low-affinity F allele have a poorer response with trastuzumab (in PFS and OR) than patients homozygous for the higher affinity V allele [50,51].

In the SOPHIA trial, Margetuximab was compared to trastuzumab in combination with chemotherapy in patients with HER2-positive metastatic breast cancer pretreated with at least two anti-HER2 regimens [52,53]. The first PFS analysis has been published in 2019 [52]. Margetuximab significantly prolonged PFS over trastuzumab (median 5.8 vs. 4.9 months; HR: 0.76, 95% CI 0.59 to 0.98; *p* = 0.033) particularly in those patients carrying a CD16A-158F allele. The benefit was observed in all chemotherapy arms, and tended to be more pronounced for patients receiving eribulin (HR: 0.66, 95% CI 0.42 to 1.05) and gembitabine (HR: 0.58, 95% CI 0.29 to 1.18). In a second interim analysis, no statistically, significant advantage was observed in terms of OS (median OS 21.6 vs. 19.8; HR: 0.89, *p* = 0.326). Overall, the combination of margetuximab plus chemotherapy was relatively well tolerated. The most common adverse drug reactions (>10 percent) with margetuximab plus chemotherapy include fatigue, gastrointestinal symptoms, headache, cough, dyspnea, infusion-related reactions, and palmar-plantar erythrodysesthesia. Additionally, left ventricular dysfunction can occur with margetuximab (1.9%). 

In total, of note, the trials leading to these drugs’ approval differed in design and no comparisons exist between them. Trastuzumab deruxtecan showed unprecedented activity in highly pretreated patients in a non-randomized trial while tucatinib was formally compared to the standard-of-care in a well-designed randomized trial. Both regimens currently represent therapeutic choices in patients pretreated with TDM-1; however, both drugs harbor significant toxicities. Margetuximab is a relatively newer agent with only modest benefits over trastuzumab and which shows an optimal safety profile [54].

Table 1 summarizes the data concerning the approved anti-HER2 molecules.

### 4.3. Other Approaches

Despite these multiple therapeutic options for patients with metastatic HER2-positive breast cancer, resistance emerges as an ineradicable problem. 

The dysregulation of the PI3K/PTEN pathway, a major downstream component of HER signaling, is a main mechanism of resistance to HER2-targeted therapies. Dysregulation, by activation, of the PI3K/AKT pathway includes activating mutations in PI3KCA and partial/complete loss of the tumor suppressor PTEN [55,56]. Several approaches including mTOR inhibition to restore trastuzumab sensitivity have been explored. The phase III trial BOLERO-3, which investigated the addition of everolimus to trastuzumab and vinorelbine in trastuzumab-resistant patients, showed a modest but statistically significant prolongation of PFS from 5.78 months to 7.0 months (HR: 0.78, 95% CI 0.65 to 0.95; *p* = 0.0067) while OS was similar in both treatment arms [57]. In this study, PTEN concentrations seemed to be a potential biomarker for everolimus efficacy. A key limitation of adding everolimus to these regimens is the significant toxicity of this drug, mainly stomatitis (67% vs. 32%) and diarrhea (56% vs. 47%). 

Another novel agent of interest in this pathway is the alpha-specific PI3K inhibitor alpelisib. It has been studied in combination with T-DM1 in a phase I study in patients with HER2-positive metastatic breast cancer resistant to trastuzumab [57]. In this study, the ORR was 43% with an acceptable tolerance profile [58]. 

The role of the immune microenvironment in modulating the tumor response to treatment has been well studied. Most trials that studied the correlation of tumor infiltrating lymphocytes (TILs) and response to HER2-targered therapy are in the chemotherapy setting, which could be a confounding factor in judging the predictive value of TILs on the response to anti-HER2 therapy. In the metastatic setting, a subanalysis of tumors from the CLEOPATRA trial revealed that higher levels of TILs correlated with improved overall survival in patients treated with anti-HER2 therapy and chemotherapy [59]. However, in the adjuvant setting, there are conflicting results. Stromal TILs are prognostically associated with recurrence-free survival in patients treated with chemotherapy alone, but not in patients treated with chemotherapy plus trastuzumab. On the other hand, high stromal TILs are predictive of lack of trastuzumab benefit [60]. 

Adding checkpoint inhibitors to standard HER2-based therapy to enhance the immune response is a new approach being investigated to re-sensitize resistant tumors to the HER2 blockade. In the KATE2 study, the addition of atezolizumab to T-DM1 was compared with T-DM1 alone [61]. A progression-free survival benefit was observed in the subgroup of patients with PD-L1–positive tumors but not in the intent-to-treat population. On the other hand, the PANACEA study (phase Ib/IIa study) evaluated pembrolizumab in combination with TD-M1 in patients with HER2-positive metastatic breast cancer resistant to trastuzumab. The main objectives in the intention-to-treat PD-L1+ population were achieved with an objective response rate and control rate disease in the order of 15% and 25%, respectively. These results are better in the subgroup of patients with PD-L1+ and TIL > 5% (objective response rate at 39% and the disease control rate at 47%), while no response was seen in the PD-L1– patient group [62].

Dysregulation of the cell-cycle, with high activity of cyclin D1 and CDK4/6, is another mechanism of resistance to the HER2-pathway blockade. The potential role of cyclin-dependent kinase 4 and 6 inhibitors has been suggested in preclinical models then in clinical studies to re-sensitize resistant tumors to the HER2 blockade. In the monarcHER trial, abemaciclib plus trastuzumab with or without fulvestrant was compared to chemotherapy plus trastuzumab in patients with HER2-positive breast cancer who have received two or more previous therapies. Abemaciclib, fulvestrant, and trastuzumab resulted in the longest median PFS (8.3 months vs. 5.7 months; HR = 0.67, *p* = 0.051), with also improved median overall survival 31.1 months vs. 20.7 months [63,64]. However, the median PFS in the abemaciclib and trastuzumab arm without fulvestrant was not significantly different from that in the trastuzumab and chemotherapy arm suggesting a crosstalk between HER2 and the cyclin signaling pathways in HR+ tumors only. The ongoing phase III PATINA study is evaluating the efficacy and safety of palbociclib added to anti-HER2 and endocrine therapy in the maintenance phase after first-line chemotherapy/anti-HER2 therapy [65].

## 5. HER2 Low: A New Disease

A new entity has recently been defined; it concerns the subpopulation of breast cancers which weakly express the HER2 receptors (HER2 immunohistochemistry 1+ or 2+, but ISH negative) and which were considered until a few years ago as HER2-negative breast cancer [66]. Novel antibody-drug conjugates targeting HER2 receptors show high activity in this entity of breast cancer.

### 5.1. Fam-Trastuzumab Deruxtecan 

In addition to their benefit in HER2-over-expressing metastatic breast cancer, fam-trastuzumab deruxtecan has shown remarkable results in patients with heavily pretreated, HER2 low advanced breast cancer. This is a phase Ib study which tested famtrastuzumab deruxtecan in 54 patients at a dose of 5.4 or 6.4 mg/kg administered intravenously every 3 weeks until disease progression or intolerable toxicity [67]. The study demonstrated an objective response rate of 37% (95% CI, 24.3, 51.3%) and a median duration response of 10.4 months.

In a randomized trial, there were 557 patients with advanced breast cancer previously treated by one or two lines of chemotherapy, and all of them had a low HER2 breast cancer. In this trial, fam-trastuzumab deruxtecan was compared to a treatment of the physician’s choice (TPC) [67]. Fam-trastuzumab deruxtecan significantly prolonged PFS over the physician’s choice group (median 9.9 vs. 5.1 months; HR: 0.50, 95% CI 0.40 to 0.63; *p* < 0.001). This benefit is also observed in overall survival; the median OS was 23.4 months versus 16.8 months (HR 0.64, 95% CI, 0.49 to 0.84). Approximately 89 percent of patients enrolled in this study had hormone receptor positive disease; the benefit of fam-trastuzumab deruxtecan was noted independently of hormone receptor expression, but there are only limited data in the subgroup of HR negative tumors. Adverse events of grade 3 or higher were noted in 52.6% of patients treated with fam-trastuzumab deruxtecan and 67.4% of those who received chemotherapy. In the experimental group, drug-related interstitial lung disease or pneumonitis was noted in 12 percent, while 0.8 percent had grade 5 events [68].

### 5.2. Trastuzumab Duocarmazine

Trastuzumab–duocarmazine (SYD985) is a novel antibody-drug conjugate in earlier phases of clinical development, which includes a humanized HER2 antibody (trastuzumab) and a DNA-alkylating duocarmycin conjugate (duocarmazine) [69]. 

In a phase I study, the effects of trastuzumab–duocarmazine in advanced breast cancer patients with a variable HER2 status and refractory to standard of care was evaluated. This study demonstrated the early signs of efficacy of trastuzumab–duocarmazine in this group of patients regardless of HER2 status [69]. In the low HER2 population (47 patients), the partial response was noted in 6 of the 15 HR-negative patients (40%, 95% CI 16.3–67.6) and 9 of the 32 HR-positive patients (28%, 95% CI 13.8–46.8) [70].

In the TULIP trial, trastuzumab–duocarmazine was compared to a treatment of the physician’s choice (TPC) in patients with HER2-positive metastatic breast cancer pretreated with at least two anti-HER2 regimens [70]. Trastuzumab–duocarmazine improved the median PFS (7 vs. 4.9 months; HR: 0.64, 95% CI 0.49 to 0.84, *p* = 0.002) and median OS (20 vs. 16.3 months) compared to TPC [71]. During the study, the most common adverse events of grade 3 or higher were ocular toxicity (21.2%) and interstitial lung disease (2.4%) [71]. In view of these results, trastuzumab–duocarmazine may provide a new treatment option for previously treated patients with a HER2-positive metastatic breast cancer.

## 6. Conclusions

Over the last decades, a better understanding of tumor biology and resistance mechanisms in HER2-positive breast cancer has favored the development of new, specific, and more potent anti-HER2 drugs, enriching the therapeutic arsenal.

Despite the various clinical trials and new therapies, the double anti-HER2 blockade has kept its place as the “gold standard” in the initial management of metastatic HER2 breast cancer. In the second line, the T-DXd has become a very interesting option because of its superiority to the T-DM1. The place of the latter has become less clear in the first therapeutic lines, especially with the advent of the new tyrosine kinase inhibitor “tucatinib” which has shown its benefit even on brain lesions. However, despite the diversity of anti-HER2 therapies, several questions remain open and still await an answer. In particular, on the biomarkers used to select responder patients and the best therapeutic combinations and sequences.

## Figures and Tables

**Table 1 cancers-15-01738-t001:** Approved Anti-HER2 Therapies in Advanced Breast Cancer.

Drug	Line	Clinical Trial	Control Arm	Route	OS Benefit	ESMO-MCBS
**Anti-her2 antibodies**						
**Trastuzumab** **+** **Pertuzumab**	First	CLEOPATRA Phase III Associated to Docetaxel	Trastuzumab + Docetaxel	IV or SC	Yes	4
**Trastuzumab** **+** **Pertuzumab**	First	PERUSE Phase III Associated to Docetaxel	Associated to Paclitaxel or Nab-paclitaxel	IV or SC	-	-
**Trastuzumab** **+** **Pertuzumab**	First (Maintenance)	PERTAIN Phase II Associated to anastrozole or Letrozol	Trastuzumab + anastrozole or Letrozol	IV or SC	-	-
**Trastuzumab** **+** **Pertuzumab**	Second	PHEREXA Phase III Associated to Capecitabine	Trastuzumab + Capecitabine	IV or SC	Yes	-
**Margetuximab**	Third or plus	SOPHIA Phase III Associated to chemotherapy	Trastuzumab + chemotherapy	IV	No	-
**Anti HER2 antibody-drug conjugate**						
**TDM-1**	Second	EMILIA Phase III	Capecitabine plus Lapatinib	IV	Yes	4
**Fam-trastuzumab Deruxtecan**	Third or plus	Destiny-Breast 02 Phase III	Treatment physician’s choice	IV	Yes	-
**Fam- trastuzumab Deruxtecan**	Second	Destiny-Breast 03 Phase III	TDM-1	IV	Yes	3
**Anti HER2 Tyrosine Kinase inhibitors**						
**Lapatinib**	Third and beyond	Lapatinib + Capecitabine	Capecitabine Alone	Oral	No	3
**Neratinib**	Third or fourth	Neratinib + Capecitabine	Lapatinib + Capecitabine	Oral	No	2
**Tucatinib**	Third and beyond	HER2CLIMB Phase III Tucatinib Capecitabine Trastuzumab	Placebo Capecitabine Trastuzumab	Oral	Yes	3

Abbreviations: ESMO-MCBS, European Society for Medical Oncology-Magnitude of Clinical Benefit Scale; IV, intravenous; OS, overall survival; SC, subcutaneous; T-DM1, trastuzumab-emtansine.

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
