# Peer review of "HER2-Positive Metastatic Breast Cancer: Available Treatments and Current Developments"

_cancers, 2023, doi:10.3390/cancers15061738_

Round 1

Reviewer 1 Report

This is a review on the treatment of HER2-positive metastatic breast cancer, providing data that has already been widely published in clinical guidelines and in the main articles of pivotal trials. I considered that for its publication in a high-impact journal such as Cancers, a review should focus on the main differences in the profile of patients included in the studies (for example, data on long-term survivors, data on cardiotoxicity, CNS disease, and rate of response etc.), about HER2 heterogeneity (from the clinical and molecular perspective) and on all of the major resistance mechanisms.

Author Response

dear reviewer
This article aims to provide an overview of antiher2 molecules that have revolutionized the management of Her2 positive breast cancer. data on brain metastases were mentioned in the article particularly for tucatinib and trastuzumab deruxtecan.
cardiotoxicity has also been mentioned. the cardiac safety data for this family of drugs is practically similar, requiring regular monitoring of cardiac function by echocardiography.

HER2 heterogeneity (from a clinical and molecular point of view) and all the major resistance mechanisms could be the subject of a separate article, in order to better understand these aspects.

Reviewer 2 Report

In this manuscript, the authors Essadi et al. have described the available treatments and current developments for HER2 positive Metastatic Breast Cancer. This review lacks originality, the information provided in this review is very much like “Current and future management of HER2-positive metastatic breast cancer”. The references are not appropriate. The authors need to cite “Current and future management of HER2-positive metastatic breast cancer” by Saez et al and “Targeting HER2-positive breast cancer: advances and future directions” by Swain et al; both have discussed this topic much more comprehensively. Thus, there is a faint scope for the readers to learn anything new from this review. I would recommend this review is not suitable for publication in this journal.

There are a few other points that require attention,

  1. To reach out to a more generalized audience the authors should provide some figures having graphical representations of the structures of the approved treatments.
  2. The authors can also include some tables or flow charts to make the review more interesting and better understandable for the reader.
  3. The authors need to weave in some expert opinions across the narrative.
  4. Use any one way of representing either HER2 or Her2.

Author Response

dear reviewer
in this article we mainly target young oncologists.

this is an overview of all the anti HER2 molecules validated in the treatment of metastatic breast cancer.

I respect your point of view, but I find that the references cited are just as respectable as those that you have proposed.

We have unified the writing form of HER2 on the text.

Following your recommendations, we have added a table that summarizes the data included in the text.

Reviewer 3 Report

The study by Ismail Essadi and collaborators is a systematic review of the different therapeutic approaches against HER2 in patients with HER2+ breast cancer. In general, the manuscript is very well structured in almost all its parts and the objectives proposed by the authors are in line with those of Cancers.

However, I recommend implementing some parts so that the manuscript is more complete and can be a very useful tool for readers.

Major revisions

The introduction deserves more attention and needs to be expanded, especially regarding the diagnosis of HER2+ carcinoma, the different membrane expressions, also making a mention of HER2-low. I recommend a recent review which the authors will find very helpful in this regard. Furthermore, considering the high value of the authors, please cite them in the text (DOI: 10.1159/000524227).

In the opinion of this reviewer, the KATHERINE study is also worth discussing in this review (DOI: 10.1056/NEJMoa1814017)

It is also advisable to insert tables that can summarize what has been described by the authors in order to make the manuscript more complete

Author Response

dear reviewer
follow-up to your recommendations:
we have added in the introduction a section on the pathological diagnosis of HER2 positive breast cancer.
we have also added the reference you suggested
we have added a table summarizing all the data concerning the molecules cited in the text

Round 2

Reviewer 2 Report

The authors need to cite “Current and future management of HER2-positive metastatic breast cancer” by Saez et al. and “Targeting HER2-positive breast cancer: advances and future directions” by Swain et al.; both have discussed this topic recently.

Reviewer 3 Report

Most requests have been made. Publication is recommended